# Building Team Medicine in the Management of CNS Metastases

**DOI:** 10.3390/jcm12123901

**Published:** 2023-06-07

**Authors:** Archit B. Baskaran, Robin A. Buerki, Osaama H. Khan, Vinai Gondi, Roger Stupp, Rimas V. Lukas, Victoria M. Villaflor

**Affiliations:** 1Department of Neurology, The University of Chicago, Chicago, IL 60637, USA; 2Health System Clinician of Neurology (Neuro-Oncology), Northwestern Medicine Regional Medical Group, Warrenville, IL 60555, USA; 3Surgical Neuro-Oncology, Northwestern Medicine Central DuPage Hospital, Winfield, IL 60190, USA; 4Department of Radiation Oncology, Nothwestern Medicine West Region, Lou & Jean Malnati Brain Tumor Institute, Northwestern University, Warrenville, IL 60555, USA; 5Neuro-Oncology Division, Neurological Surgery, Medicine (Hematology and Oncology), Neurology, Department of Neurology, Lou & Jean Malnati Brain Tumor Institute Northwestern University, Chicago, IL 60611, USA; 6Neuro-Oncology Division, Department of Neurology, Lou & Jean Malnati Brain Tumor Institute, Northwestern University, Chicago, IL 60611, USA; 7Department of Medical Oncology and Therapeutics Research, City of Hope, Duarte, CA 91010, USA

**Keywords:** brain metastases, delivery of health care, leptomeningeal metastases, tumor board

## Abstract

CNS metastases are often terminal for cancer patients and occur at an approximately 10-fold higher rate than primary CNS tumors. The incidence of these tumors is approximately 70,000–400,000 cases annually in the US. Advances that have occurred over the past two decades have led to more personalized treatment approaches. Newer surgical and radiation techniques, as well as targeted and immune therapies, have enanled patient to live longer, thus increasing the risk for the development of CNS, brain, and leptomeningeal metastases (BM and LM). Patients who develop CNS metastases have often been heavily treated, and options for future treatment could best be addressed by multidisciplinary teams. Studies have indicated that patients with brain metastases have improved survival outcomes when cared for in high-volume academic institutions using multidisciplinary teams. This manuscript discusses a multidisciplinary approach for both parenchymal brain metastases as well as leptomeningeal metastases implemented in three academic institutions. Additionally, with the increasing development of healthcare systems, we discuss optimizing the management of CNS metastases across healthcare systems and integrating basic and translational science into our clinical care to further improve outcomes. This paper summarizes the existing therapeutic approaches to the treatment of BM and LM and discusses novel and emerging approaches to optimizing access to neuro-oncologic care while simultaneously integrating multidisciplinary teams in the care of patients with BM and LM.

## SUMMARY TABLE


The treatment of brain metastases (BM) and leptomeningeal metastases (LM) includes a combination of surgical intervention, radiation therapy, systemically administered therapies, and therapies directly administered to the CNS.

Surgical interventions for BM include craniotomy and emerging modalities such as laser interstitial thermal therapy (LITT), which are often utilized with large tumor metastases that produce neurologic symptoms and increase intracranial pressure; for LM, options are more limited to the placement of intraventricular reservoirs for intrathecal chemotherapy or shunt placement for communicating hydrocephalus.

Radiation therapies for BM include whole-brain radiation therapy (WBRT) and stereotactic radiosurgery (SRS), of which neurocognitive toxicity remains a pressing concern; for LM, options include WBRT and craniospinal irradiation (CSI).

Systemically administered therapies for BM include tyrosine kinase inhibitors (TKI), such as osimertinib and tucatinib, and immune checkpoint inhibitors (ICI), such as cytotoxic T-lymphocyte-associated antigen 4 (CTLA4) inhibitors and programmed death 1 (PD1) pathway inhibitors; for LM, there is not a standard-of-care systemic therapy, with overall prognosis remaining poor.

Intrathecal (or intra-CSF) chemotherapy is an ongoing area of study for the treatment of LM, with novel therapies demonstrating some promise.

Hospital-system-wide tumor boards, also known as multidisciplinary cancer meetings (MCM), are emerging settings through which providers representing multidisciplinary neuro-oncology subspecialties meet to discuss complex cases and facilitate information sharing for the improvement of patient outcomes. These tumor boards may be in person or virtual, which remains an ongoing area of interest.

Telemedicine has been a growing technology since COVID-19, with widespread potential for improving access and quality of care in the field of neuro-oncology.


## 1. Background

In this review, we discuss the clinical management of brain metastases (BM) and leptomeningeal metastases (LM), which are diseases managed by a multidisciplinary team of medical oncologists in conjunction with neuro-oncologists, radiation oncologists, and neurosurgeons [1]. There remains an unmet need to provide specific, central nervous system (CNS)-metastases-directed treatment in metastatic solid tumors. A multidisciplinary-team-based medicine model may add value for patients, practitioners, and hospital systems following such an approach. For example, the survival outcomes in brain metastases patients cared for at academic institutions appear superior [2]. This benefit of care in high-volume centers is in line with what has been observed for patients with glioblastoma [3,4]. While numerous non-clinical factors, including selection bias, may contribute to this benefit, we propose that specific factors related to the coordinated multispecialty oncology care model may positively impact patient outcomes. For example, a multidisciplinary approach has been shown to decrease treatment-associated clinical encounters and decrease the time to the initiation of radiation therapy for patients with BM [5]. However, available data to indicate which are the important elements and how to structure a coordinated, interdisciplinary care model are sparse and often lack granularity [6].

The various therapeutic modalities chosen and delivered by assorted sub-specialists in the care of patients with BM and LM will be briefly reviewed. These interventions include surgical intervention, radiation therapy, systemically administered therapies, and therapies directly administered to the CNS. How individual physicians can use these therapeutic modalities within a team model and in the context of a multidisciplinary care plan developed across a healthcare system will be highlighted. Finally, the integration of basic and translational science discoveries into the clinical management of BM and LM points toward future areas of focus to continue to benefit this complex oncology patient population.

## 2. Multi-Disciplinary Clinical Management of Brain Metastases

### 2.1. Epidemiology and Prognosis of Brain Metastases

The exact incidence of BM from solid tumors is not as certain as that of primary CNS tumors. It is thought there are ~70,000–400,000 new cases in the United States per annum. If correct, this makes them ~10-fold more common than primary CNS malignancies [1,7]. In turn, the scope of the problem presented by BM is substantial. Coordinated, high-touch multidisciplinary management may be essential for all patients with BM. This may be the case when the burden of progressive systemic disease supersedes CNS disease in driving morbidity and mortality, a concept that can be quantified as brain metastases velocity. Patients who develop progressive BM at a rapid annualized rate are at greater risk of dying from CNS disease [8,9]. Logistical and organizational factors also impact the feasibility of team-based multidisciplinary care for BM patients, including a limited workforce of sub-specialty providers, BM-specific physical and cognitive patient debility, and the complexities presented by health system infrastructure for well-coordinated care.

Until recently, BM was viewed as being associated with a universally dismal prognosis [10]. We now have a more nuanced perspective of outcomes for these patients. For patients with BM from non-small cell lung, breast, melanoma, gastrointestinal, and renal cancers, median survival ranges from 7–47 months, 3–36 months, 5–34 months, 3–17 months, and 4–35 months, respectively per the Disease Specific Graded Prognostic Assessment (DS-GPA). Numerous factors comprise GPA score, including histology, molecular characteristics, patient age, performance status, and the number of metastases [11]. Considering these various points is important when developing an optimal plan tailored to specific patient characteristics and circumstances.

### 2.2. Surgical Therapies

Surgical resection has demonstrated efficacy in prolonging survival in patients with solitary BM. The role of surgical resection is predominantly in patients with large BM that produce either neurologic symptoms or those associated with increased intracranial pressure due to mass effect. It also serves a diagnostic role, including when a patient may benefit from more detailed molecular characterization of their BM. The role of surgical resection in oligometastatic disease is less clear, although it can be considered when there is a dominant symptomatic BM in the setting of other smaller lesions. One key study was the pivotal phase 3 randomized controlled trial comparing the effectiveness of surgical resection plus post-operative whole-brain radiotherapy (WBRT) vs. WBRT alone. This trial demonstrated that surgical resection followed by radiotherapy was superior to WBRT for the treatment of a solitary BM. The median overall survival (OS) was over 15 months in the surgical group vs. over 6 months in the RT-only group (*p* < 0.0009) [12]. Superiority was also observed across numerous other clinically relevant endpoints.

MRI-guided laser interstitial thermal therapy (LITT) is a minimally invasive surgical alternative for metastases up to 3 cm in diameter in locations where surgical resection may not be readily feasible [13]. The technology delivers laser light through a stereotactically navigated fiber optic probe to create thermal damage, leading to cellular death within the target lesion [14]. In patients with BM and radiation necrosis, who may be sicker and thereby unable to tolerate resection of the necrotic lesion, LITT may present itself as a viable alternative; in patients with radiation necrosis, it has been shown to decrease lesion size and improve progression-free survival [15]. Some studies suggest that LITT may even be superior to bevacizumab in the management of select patients with radiation necrosis [16]. Similarly, other studies have suggested comparable outcomes between LITT and other surgical resection approaches for patients with brain metastases [17]. However, to date, the evidence comparing LITT to craniotomy or medical management for the management of radiation necrosis or brain metastases is largely retrospective (e.g., LITT vs. craniotomy [18], LITT vs. bevacizumab [16]), with one multicenter prospective study demonstrating benefits to laser ablation following SRS in patients with brain metastases and radiation necrosis [19]. One additional phase I clinical trial is underway, exploring the efficacy of a combination therapy of LITT with Pembrolizumab for recurrent BM (NCT04187872).

### 2.3. Radiation Therapies

Radiation therapy (RT) has formed the backbone of the treatment of BM. The most common approaches, WBRT and stereotactic radiosurgery (SRS), have different benefits in terms of acute and late toxicity, ease of implementation, and treatment duration, without clear differences in survival from CNS failure in oligometastatic disease (OMD). OMD has been defined as up to five metastatic lesions [20]. Prophylactic cranial irradiation (PCI) is another therapeutic approach using WBRT that has demonstrated success in reducing the incidence of symptomatic BM in patients with limited- and extensive-stage small cell lung cancer (ES-SCLC), as well as improving OS [21]. However, significant knowledge gaps surrounding its neurotoxicity and unique effects in different subgroups of patients have made its continued routine use somewhat controversial [22,23]. With ongoing investigations evaluating the growing role of SRS in this disease, it is possible that this current approach may be supplanted.

WBRT has the advantages of ubiquity in being able to be implemented quickly, its ability to be administered across healthcare systems with varying resources, and the ability to treat both macro- (i.e., radiographically visible) and micrometastatic CNS disease. A key limitation of WBRT is the neurocognitive toxicity from irradiating healthy brain tissue unaffected by tumors and the resulting deterioration of patient independence and quality of life (QOL) [24]. In contrast, SRS can provide local control of BM without the risk of the potential late neurotoxicity associated with larger volumes of radiation and at the expense of not treating micrometastatic disease.

There have been efforts to avoid the neurocognitive toxicity of WBRT [25]. Two recent interventions have been demonstrated to reduce the risk of delayed cognitive decline: (i) the addition of the N-methyl-D-aspartate receptor (NMDAR) antagonist memantine, a drug marketed to treat Alzheimer’s disease, and (ii) hippocampal avoidance WBRT (HA-WBRT) [26], i.e., a careful method of treatment planning that protects the hippocampi from high-dose radiation exposure when delivering WBRT [26]. The addition of 6 months of memantine to WBRT in the randomized RTOG 0614 trial resulted in delayed cognitive decline in the investigational arm [27]. The phase 3 NRG CC-001 trial used this same approach and investigated the addition of HA-WBRT. The trial compared the neurocognitive decline in patients with BM treated with either standard WBRT or HA-WBRT [28]. The employment of HA-WBRT was associated with lesser deterioration of executive function as well as learning and memory without any detriment to overall survival or intracranial progression-free survival. This benefit was observed across all patient ages and was first noted at 4 months, persisting out to at least 12 months.

There has been continual work on optimizing the role of SRS for BM. SRS has been shown to be both safe and efficacious for treating several “oligo” BM. Absolute tumor number, cumulative tumor volume, and tumor location are all factors that may influence the efficacy, safety, and tolerability of SRS for BM, making the discussion more complex than simply defining an optimal numeric cut-off for the number of brain metastases. In patients with 1–3 brain metastases, SRS has already replaced WBRT as the standard of care [29]. A more recent randomized phase III trial demonstrated level 1 evidence for the superiority of SRS over WBRT in non-melanoma patients with 4–15 brain metastases [30], providing support for the expansion of the number of BM that can feasibly be treated. The latest ASCO-SNO-ASTRO guidelines recommend SRS for patients with 1–4 unresected, asymptomatic brain metastases, excluding small cell lung cancer (SCLC) [31]. Two additional phase III clinical trials (NRG Oncology CC009 and NRG BN009) are currently underway. NRG-CC009 is comparing SRS to HA-WBRT plus memantine for 10 or fewer BMs from small cell lung cancer with a primary endpoint of cognitive function failure; NRG BN009 is comparing salvage SRS to salvage HA-WBRT for distant brain relapse following upfront SRS with high brain metastasis velocity (BMV). BMV is defined as the total number of new brain metastases a patient develops over time since their first treatment with SRS; it is a newer metric that has been associated with neurologic death and overall survival and is a predictor for determining a patient’s need for salvage WBRT after initial distant brain failure following upfront SRS alone [9].

For large BM and limited intracranial disease burden, post-operative SRS to the surgical resection cavity may be employed as a component of multimodal therapy [32]. Over time, prospective trials have validated the superiority of post-operative SRS over WBRT for this indication [33,34]. However, post-operative SRS has been associated in some studies with radiation necrosis, leptomeningeal metastatic spread, and local failure at the SRS treatment site, thereby prompting the exploration of the use of pre-operative SRS [35]. One retrospective multi-institutional analysis of 279 patients found lower rates of radiation necrosis, leptomeningeal spread, and local failure in patients undergoing multimodal therapy for large BM with limited intracranial disease [36]. A recently initiated phase III trial is exploring the efficacy of pre-operative versus post-operative SRS for patients with surgically resected BM (NRG BN012; NCT05438212). Currently, the ASCO-SNO-ASTRO guidelines recommend SRS to the surgical cavity in patients with 1–2 resected BM and SRS vs. WBRT vs. combination therapy for other patients [31].

### 2.4. Systemic Therapies

Traditionally, systemic chemotherapy had a limited role in the management of BM due to the CNS penetration of available agents, yet growing evidence suggests that systemically effective therapies of newer small molecules are also demonstrating responses in the CNS. The intracranial failure rate has been reduced in randomized trials, likely impacted by the low CNS penetration of the agents used. A wide range of therapeutic agents are now considered in the management of BM. The most frequently used agents are systemic targeted therapies (e.g., tyrosine kinase inhibitors (TKI)) and immune checkpoint inhibitors (ICI).

One example within the targeted therapy class of systemic therapy is the TKI osimertinib, a third-generation agent for the treatment of epidermal growth factor receptor (EGFR) mutant non-small cell lung cancer (NSCLC). Osimertinib has demonstrated efficacy in the treatment of T790M-positive advanced NSCLC BM [37]. Another TKI is tucatinib, a human epidermal growth factor receptor 2 (HER-2) inhibitor often used for the treatment of HER2(+) breast cancer. A post hoc exploratory analysis of patients with BM treated on the HER2-CLIMB trial with tucatinib in combination with capecitabine and trastuzumab further demonstrated a role in preventing intracranial disease progression, showing a 68% reduced risk of intracranial progression or death versus capecitabine and trastuzumab alone [38]. Similar results have been seen for a variety of other solid tumor types. BRAF/MEK targeted therapies, such as dabrafenib and trametinib, have become the standard of care for the treatment of melanoma brain metastases with a BRAF-MEK pathway driver mutation [39] and currently have histology-agnostic approval for tumors harboring the *BRAF* V600E mutation. Retrospective cohort data have demonstrated the efficacy and acceptable safety profiles of another TKI, cabozantinib, for the treatment of renal cell carcinoma (RCC) BM [40].

Drugs within the ICI class of systemic therapies used for the treatment of BM often target the cytotoxic T-lymphocyte-associated antigen 4 (CTLA4) and programmed death 1 (PD1) pathways. Theoretical benefits of ICI therapies include their lower toxicity profiles compared to traditional chemotherapeutic agents as well as their potential for efficacy both outside of and within the CNS [41]. Combination ICI therapies have also often been investigated for the treatment of BM. One phase II trial exploring the combination of nivolumab with ipilumab for patients with mainly asymptomatic melanoma BM demonstrated a response rate of 57% (95% CI 47–67%) and a three-year intracranial PFS of 54% (95%CI 43–64%) in 101 patients with asymptomatic melanoma BM [42]. Several ongoing trials are further investigating the incorporation of ICI for the treatment of BM secondary to multiple primary solid tumors (NCT05704647, NCT04511013, NCT05012254, NCT03873818, NCT04187872, NCT05341349, and NCT04348747).

## 3. Multi-Disciplinary Clinical Management of Leptomeningeal Metastases

### 3.1. Epidemiology and Prognosis of LM

Leptomeningeal spread is increasingly observed in advanced metastatic cancer, with a relative increase as systemic treatments and disease control have substantially improved over the last decade. LM occurs in approximately 5% of advanced solid tumors, yet autopsy series reveal a prevalence of up to 20% for asymptomatic or undiagnosed LM [43,44]. The observed incidence of LM diagnoses may be increasing due to advances in the diagnosis of LM and treatment strategies that prolong survival from systemic disease [45,46]. While cancer prognoses have broadly improved, the prognosis of LM remains poor, with survival times typically quoted at only 2 to 3 months [47,48]. The most important (and consistent) prognostic factor across numerous studies remains the patients’ performance status [49,50]. This serves as a readily assessable quantifiable measure that can help guide the aggressiveness of the management approach.

Recently, progress has been made in the understanding of patients at risk for and the pathogenesis of LM [51,52,53]. The presence of a blood–CSF barrier (distinct from the blood–brain barrier (BBB)) presents unique therapeutic challenges for the management of LM compared to BM. The blood–CSF barrier is the space between the choroid plexus and the CSF. Unlike the BBB capillaries, which form the endothelial layer of the brain parenchyma to separate brain interstitial fluid from the blood, the choroid plexus forms the epithelial layer to separate CSF from the blood; it has no tight junctions, contains fenestrations, and thereby utilizes distinct active transport mechanisms such as bulk CSF flow and transcapillary exchange to determine the concentration of molecules in the CSF [54]. Given its relative leakiness compared to the BBB, the blood–CSF barrier permits the crossing of certain substances that would not otherwise cross the BBB [55]. LM have been shown to upregulate complement component 3 (C3), activating the C3a receptor on the choroid plexus, thereby disrupting the blood–CSF barrier and enabling the passage of growth factors such as amphiregulin into the CSF that enable LM to adapt and spread in the CSF microenvironment [56]. This preclinical work raises the opportunity for further study in inhibiting C3 signaling pathways to therapeutically manipulate the blood–CSF barrier to improve the penetration of systemic chemotherapeutic agents.

### 3.2. Surgical Therapies

The role of surgical intervention in LM is limited to the placement of an intraventricular reservoir (Ommaya, Rickham) to facilitate the delivery of intrathecal chemotherapy or the placement of a shunt in patients with communicating hydrocephalus due to flow obstruction by metastatic cells in the cerebrospinal fluid [57,58]. One critical consideration for safety in proposing direct IT chemotherapy is establishing unobstructed CSF circulation flowing through the ventricular system. Chemotherapy can be delivered via lumbar puncture; an intraventricular reservoir allows not only for less burdensome repeat drug delivery but also appears to be associated with superior survival when compared to delivery via lumbar puncture based on a small retrospective series [59]. Although not as common, however, placement of a reservoir in continuity with a ventriculoperitoneal shunt with the ability to temporarily turn “off” the shunt provides an opportunity to deliver therapy and divert CSF in the same system, thereby avoiding multiple surgeries or procedures (recurrent lumbar punctures).

### 3.3. Radiation Therapies

Radiation has been used to treat LM, although practice patterns vary between institutions [48]. Different approaches for RT in this setting include focal radiation to address symptomatic bulky or obstructive leptomeningeal metastases, WBRT to treat a substantial but incomplete portion of the target space, and full craniospinal irradiation (CSI).

The benefits of CSI have traditionally been limited by significant potential CNS and extra-CNS toxicity. However, in comparison to standard photon radiation techniques, the application of protons for CSI may be a means to overcome some of the toxicities, particularly myelosuppression and radiation esophagitis. Proton RT is a type of radiation treatment [60]. A recent phase I study of proton CSI for LM patients demonstrated a favorable safety profile with self-resolution of any dose-limiting toxicities and improved overall survival compared to historical norms [61], and a subsequent small randomized phase II trial supported the superiority of proton CSI (both in terms of increased meaningful overall survival as well as reduced neurologic symptom burden) when compared to involved field photon radiation; this is the only intervention trial to have ever shown an overall survival benefit in LM [62]. This concept requires further validation, and a larger multi-institution phase 3 trial is currently in development. Another phase I clinical study is currently underway exploring an intrathecal rhenium-186 nanoliposome for the treatment of LM (NCT05034497).

### 3.4. Systemic Therapies

The role of systemic therapies in the treatment of LM is evolving. One study compared the efficacy of systemically administered high-dose methotrexate (HD-MTX) with IT MTX for patients with LM, finding that systemic MTX may be superior for the cytologic clearing of CSF and for OS in patients (mOS 13.8 months for the IV HD-MTX group vs. 2.3 months for the IT-MTX group, *p* = 0.003) [63]. However, more recent case series have demonstrated the potential viability of integrating HD-MTX into broader multimodal treatment plans for LM [64].

Following up on prior data demonstrating the benefit of combination therapy with tucatinib and capecitabine plus trastuzumab for parenchymal BM, a randomized phase II study is currently ongoing to further explore this treatment combination in LM (NCT03501979). Furthermore, a phase II trial of ANG1005, a novel taxane derivative designed to penetrate the blood–brain and blood–CSF barriers more effectively, showed that 79% of patients with LM had intracranial disease control with a median OS of 8 months [65]. Despite these findings, there is no standard-of-care systemic therapy in this LM setting, with the overall prognosis remaining poor [66].

### 3.5. CSF-Administered Therapies

The direct administration of antineoplastic agents into the CSF is a means to circumvent the blood–CSF barrier [67]. Therapeutic trends investigated within the context of clinical trials have recently been reviewed [68].

Intrathecal (or intra-CSF) chemotherapy may be delivered via surgically implanted ventricular Ommaya reservoirs or via lumbar puncture [69]. The theoretical benefit of direct intrathecal delivery is the ability of agents to achieve homogenous distribution within the subarachnoid space [70]. A recent review of intrathecal therapies for LM over a 39-year period showed that the most commonly administered intrathecal regimens consisted of a combination of singular therapy of methotrexate (MTX), cytarabine (Ara-C), and thiotepa, with the additional limited use of topotecan, pemetrexed, and systemic antibodies or immunotherapies such as trastuzumab and interleukin-2 (IL-2) [68,71,72]. The rationale for choosing each agent depends largely on their pharmacokinetic properties, namely their half-life, clearance from the CSF space, and lipophilicity [68,69]. An additional agent, sustained-release liposomal cytarabine (DepoCyt), was initially utilized but discontinued in 2017 following increased adverse events secondary to neurotoxicity [73]. Ultimately, while there are no specific guideline-based therapies for determining which intrathecal agent to use for which primary tumor, data suggest a benefit of (1) MTX in LM due to solid neoplasms, (2) trastuzumab for LM due to HER2-positive malignancies, and (3) pemetrexed for LM secondary to lung adenocarcinoma [68].

Of particular interest has been the use of antibodies for specific targets, such as HER2(+). These have proven to be very well tolerated and associated with favorable survival outcomes [74,75]. One phase I/II trial of IT trastuzumab for patients with HER2-positive breast cancer patients demonstrated an mOS of 10.5 months for the HER2-positive breast cancer population with LM versus 3.3–4.4 months for historical controls [49,74]. Another phase II study exploring IT trastuzumab for HER2-positive breast cancer demonstrated a mOS of 7.9 months [75]. Numerous other small single-arm studies of a variety of IT approaches have been explored, with modest results observed thus far [62,63,64].

## 4. Optimizing Management of CNS Metastases across a Health Care System (and Beyond)

### 4.1. Tumor Boards

The integrated, interdisciplinary team management of BM and LM necessitates seamless cross-coordination between different subspecialty providers. One such platform that has enabled this includes hospital-system-wide tumor boards, also known as multidisciplinary cancer meetings (MCM), specifically directed for CNS metastases. These can bring together experts from medical oncology, neurosurgery, neuro-oncology, radiation oncology, neuroradiology, neuropathology, epilepsy neurology, and cancer genetic counseling to review complex cases.

In oncology, tumor boards have been established as a means of improving accuracy in diagnosis, bettering adherence to clinical guidelines, advancing the integration of novel research and clinical trials in clinical management, and improving patient outcomes [76]. Prior studies have demonstrated that multidisciplinary tumor boards improve both the quality of medical services offered to patients as well as OS rates [77,78,79,80]. One retrospective study of an institution’s neuro-oncology tumor board at a large tertiary care academic medical center in Italy suggested that the multidisciplinary management of challenging cases improved overall effectiveness in managing brain tumors [76]. At this institution, complex cases often involving gliomas and brain metastases were discussed on a weekly basis between neurosurgeons, neuro-oncologists, neuro-radiologists, neuropathologists, and other key multidisciplinary teams; neuro-oncologists presented most cases, with neuroradiologists providing interpretations of possible image reassessments, with ongoing cross-team communication between all providers on updated therapies utilized for each patient’s care along with interval reassessments of previous exams [76]. The discussions enabled not only the broadening of differential diagnoses but also treatment plan changes [76]. Additional benefits described in the literature of institutions with neuro-oncologic multidisciplinary tumor boards included improved resident education, increased clinical trial access for patients, and increased published guideline adherence [81].

### 4.2. Telemedicine

Since the COVID-19 pandemic, telemedicine has rapidly expanded across the globe to facilitate both more seamless interdisciplinary care and increased access to oncologic care, particularly within neuro-oncology [82]. One review posits that certain types of neuro-oncology encounters, including chemotherapy monitoring visits, chemotherapy consent and education, second opinion visits, or discussions around new laboratory or imaging results, may all prove viable in a telemedicine setting [83]. One study of an institution’s telemedicine program for neuro-oncology visits demonstrated equivalent patient satisfaction, with time and cost savings for patients [84]. A review of the telemedicine programs at Barrow Neurological Institute and Geisinger Health during the early COVID-19 pandemic period in 2020 demonstrated that the neuro-oncology divisions were able to return to 90% or greater capacity within 6 weeks of initial closure due to both systems’ effective implementation of telehealth programs [85]. Although further research is needed, the promise of telehealth for patients and providers is the largest when considering its potential to expand access to care in remote areas, particularly for patients with financial or transportation barriers to care.

Furthermore, the ease of telemedicine platforms in enabling multidisciplinary discussions regardless of geographic location may help facilitate virtual tumor boards in the future [82]. Recently, multi-institutional virtual tumor boards (VTBs) have been an emerging platform for case discussions; a review of three VTBs in the United States demonstrated that neuro-oncology VTBs provide for faster expert analysis of clinical cases with an average response time under 24 h [86]. Thus, VTBs represent an effective means of conducting multidisciplinary care for patients with neuro-oncologic disease burdens without the inhibition of institutional barriers.

## 5. Integration of Science into Clinical Care

The molecular pathological characterization of CNS tumors is becoming increasingly important in diagnosis and clinical management [87]. Practically, there may be greater value in performing molecular pathology evaluations earlier versus later in a patient’s treatment course. The application of next-generation sequencing to detect mutations and oncogenic fusions can also be applied to brain metastases [88]. Genomic analysis of BM has compared them to systemic tumor metastases to distal extracranial and regional lymph node sites, finding that BM harbor several distinct genomic alterations compared to primary tumors, particularly in the CDK and PI3K/AKT/mTOR gene pathways [89]. As such, BM may demonstrate sensitivity to inhibitors targeting these pathways when those aberrancies are present. An ongoing National Cancer Institute (NCI) cooperative group phase II randomized clinical trial (NCT03994796) is investigating agents with known CNS/blood–brain barrier penetrance, including abemaciclib, paxalisib and entrectinib, to selectively target these pathways, as well as other pathways, such as NTRK and ROS1. In this trial, 136 patients with new, recurrent, or progressive BM will be divided into three experimental arms: the first arm with CDK mutations, the second with PI3K mutations, and the third with NTRK/ROS1 mutations. The groups will, respectively, be administered twice daily oral abemaciclib, daily oral paxalisib, and daily oral entrectinib, each for a total of 28 days, with cycles repeating every 28 days. The primary outcome measure will be the objective response rate in the brain as determined by Response Assessment in Neuro-Oncology (RANO) criteria; the secondary endpoints will consist of systemic response and clinical benefit rates, duration of response, PFS, OS, and adverse rate incidence. The goal of the trial is to determine the efficacy of targeted therapies in patients harboring specific BM gene mutations. Although this targeted, personalized approach for patients with BM will require tissue from the CNS for analysis, it may represent a new paradigm for the clinical management of new, recurrent, or progressive BM.

## 6. Conclusions

The management of CNS metastases is an important component of oncologic care. It is optimal if this is integrated into the other aspects of the care of this patient population. This may be complex as some patients benefit from surgical, radiation, systemic, and/or CNS-delivered therapies delivered by a panoply of subspecialty providers. There are numerous models on how to best accomplish this high-quality multi-disciplinary care, with no single approach demonstrating definitive superiority over others. Considerations for implementation include resources, including sub-specialty clinical care providers with adequate availability and interest. We present a broad overview of how this care can be delivered.

## Data Availability

Not applicable.

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
