# Peer review of "Building Team Medicine in the Management of CNS Metastases"

_jcm, 2023, doi:10.3390/jcm12123901_

Round 1

Reviewer 1 Report

This is a manuscript arising from a tertiary academic medical center that describes multidisciplinary management of parenchymal and leptomeningeal brain metastases and how various modalities are currently employed and where management may be improved in the future through further collaboration and application of evidence-based care. 

General Comments:  

This review is comprehensive in its treatment of the subject matter. It is relevant to oncology because of the prevalence of brain metastases and leptomeningeal metastases from solid tumors and the potential for improvement of patient care through assessment of outcomes with well-designed clinical trials and application of evidence-based care. There are areas that need further explication in my opinion to add value to this contribution. The discrimination of focal leptomeningeal disease from disseminated leptomeningeal disease in relation to the employment of therapies that may be appropriate is insufficient. Should a patient with focal leptomeningeal disease have an Ommaya placed? Should she get craniospinal irradiation with protons? Should it be treated with radiosurgery? Is high-dose methotrexate the most appropriate option? Also, where clinical trials are needed to help resolve management controversies that recurrently develop should be pointed out to provide guidance for readers to areas where their contribution may help answer questions.

Specific Comments: 

  1. L94: Please add the word ‘median’. 

  1. L119: Consider ‘necrotic’ for ‘radiation-necrosed’. 

  1. L137: This is one group’s definition of oligometastatic; can a case be made that this is a generally accepted definition? Different specialties dealing with brain metastases have different opinions on this subject (doi: 10.3171/2010.8.GKS10999). Suggest: ‘has been’ rather than ‘is generally’. 

  1. L138: Aren’t all brain metastases treatable with WBRT? This doesn’t make sense as a definition criterion. Please clarify. 

  1. L202: Consider adding ‘low’ before ‘CNS’. 

  1. L204-5: I do not understand this sentence. 

  1. L215: Are the words ‘another TKI’ needed in this location in this sentence? 

  1. L220: Please add the term ‘with a BRAF-MEK pathway driver mutation’ to this sentence, or rephrase to indicate that these drugs are appropriate for a subset of melanoma patients. 

  1. L223: It may be worth adding the mechanism of action (‘another TKI’) to the sentence about cabozantinib. 

  1. L236-243: You include two CD-19 CAR T-cell trials. What relevance do these trials have for brain metastasis and leptomeningeal metastasis management? I would delete them for lack of pertinence. 

  1. L268: What is the importance of this fact about C3-C3a to the management of leptomeningeal brain metastases. It’d be nice to tie it in (rather than just dropping it in there) to clinical management in some way. 

  1. L279: CSF does not flow throughout the brain and spinal cord. 

  1. L284: Suggest ‘temporarily’. 

  1. L294: Suggest ‘some of’ before ‘the toxicities’. 

  1. L295: This sentence introducing proton beam therapy is non-informative. 

  1. L309: Please add ‘systemically administered’ before ‘high dose’. 

  1. L310: It may be possible to delete ‘the’. 

  1. L316: Consider switching ‘the’ for ‘this’. 

  1. L326: I do not think ‘Intra-CSF’ needs to have the initial ‘I’ capitalized. 

  2. L329: I am unfamiliar with the term ‘scoping review’. 

  3. L334: I think that this sentence needs a verb. 

  1. L340: Please add the word ‘positive’ or a ‘+’ sign after ‘HER2’. 

  1. L448: ‘Elsevier’ is misspelled. 

  1. L449: There needs to be a period following the word ‘CranioVation’.

The authors should closely read their manuscript contributions to make sure that clarity and concision are being achieved with what they choose to submit.

Reviewer 2 Report

This is a very relevant and nicely written summary of the current management of brain metastases and leptomenigeal disease (neoplastic meningitis), similar to other reports (i.e., Neuro-Oncology, Volume 24, Issue 10, October 2022, Pages 1613–1646).

The reader has to see the novelty of this very delightful review paper, I suggest the authors to write a few sentences or paragraph to understand the originality of their manuscript. 

The term leptomeningeal disease is usually used to describe the meningeal involvement by primary tumor spread; nevertheless, the term leptomeninges refers to the pia and arachnoid layers only, leaving the dura and arachnoid space aside. Neoplastic meningitis, on the other side, refers to the involvement of the leptomeninges, the pachymeninges, and the arachnoid space.
